# Recurrent Urinary Tract Infections (UTIs): A Review and Proposal for Clinicians

**DOI:** 10.3390/antibiotics14010022

**Published:** 2025-01-02

**Authors:** Dino Sgarabotto, Elena Andretta, Camilla Sgarabotto

**Affiliations:** 1Policlinico di Abano Terme, 35031 Abano Terme, Padova, Italy; 2Spinal Unit, Oras Hospital, 31045 Motta di Livenza, Treviso, Italy; kirafluss@gmail.com; 3Department of Anesthesia and Intensive Care Unit, ULSS6 Euganea, 35013 Cittadella, Padova, Italy; camilla.sgarabotto@gmail.com

**Keywords:** recurrent urinary tract infections (UTIs), bacterial persistence, intracellular bacterial communities (IBCs), pulsed antibiotic therapy, intracellular active antibiotics

## Abstract

The pathogenesis of recurrent urinary tract infections (rUTIs), a common problem in the female population, is becoming better understood following recent studies of bacterial persistence and intracellular bacterial communities. Incorporating these new insights, we propose pulsed antibiotic therapy with intracellular activity as a possible treatment for rUTIs.


*“Felix qui potuit rerum cognoscere causas” Virgil, Georgics, II, 489*



*“Fortunate, who was able to know the causes of things”*


## 1. Introduction

Recurrent urinary tract infections (UTIs) are defined as two infections in a 6-month period, or three infections over 12 months, with complete resolution for at least 2 weeks. Clinical features of UTIs most commonly consist of frequent urination, urgency, dysuria, strangury and bad smelling urine. Up to 75–90% of UTIs are due to *Escherichia coli* [1,2]; other usual pathogens are *Klebsiella*, *Proteus*, *Enterococcus* and *Staphylococcus*.

Recurrent UTIs represent a common problem in the female population: the cause of the infection has been considered mainly as reinfection with a new bacterial type and not a consequence of bacterial relapse with the same strain as the initial infection. In cases of relapse with the same bacterium species, a radiologic evaluation is appropriate in order to evaluate the presence of stones, unilateral infected atrophic kidney, urethral abnormalities and other anatomical anomalies [3].

Currently, it is well established that recurrent UTIs are often due to the very same bacterium that remains hidden in the bladder between an attack of cystitis and the following infection that is undetected by urine culture collected 2 or 3 weeks after a short antibiotic cycle. A positive urine culture has a cut-off between 10^4^ and 10^5^ CFU/mL depending on specific conditions (acute cystitis or asymptomatic bacteriuria) and the type of bacteria (Gram+ or Gram−); therefore, a lower quantity of bacteria can go undiscovered [4].

Identical bacteria hidden in the bladder during intermittent cystitis may lead to bacterial persistence, which is still not completely understood, despite being studied for the past 15 years. Intracellular bacterial communities (IBCs) in the urothelial cells of the bladder mucosa may be a further explanation for cystitis reoccurrence.

## 2. Bacterial Persistence

Bacterial persistence is the phenomenon of an identical pathogen remaining dormant between two isolated episodes of UTIs. It is more than 70 years that the concept of bacterial persistence has been distinguished by bacterial antibiotic resistance. However, the existence of bacterial persistence does not yet play a part in clinical practice even if there is sound evidence of its existence [5]. Indeed, Figure 1 represents the occurrence of a fraction of persistent cells within a bacterial population undergoing antibiotic therapy (panel a): persistent cells (blue) are phenotypic variants of the wild type (antibiotic-sensitive bacterial cells in green) that are formed at low frequency in the population (panel b). Upon antibiotic treatment, the majority of cells are killed off, leaving the tolerant bacterial cells unaffected (panel c). Removal of the antibiotic pressure allows the persistent cells to resume growth (panel d), hence remaining susceptible to the antibiotic.

Therefore, antibiotic resistance is not the only mechanism employed by bacteria to survive antibiotic exposure. Unlike resistance, which allow the resistant bacteria to grow, the persistent bacteria do not show any growth during antibiotic exposure and do not rely on the minimum inhibitory concentration or inactivation of the antibiotic itself.

## 3. In Vitro Studies

Bacterial persistence under antibiotic pressure determines a phenotypic change in a small proportion of bacteria that is transient and reversible. Persistence is not a passive phenomenon, rather it is regulated by specific molecular mechanisms.

### 3.1. Toxin-Antitoxin

Bacterial toxin–antitoxin (TA) is composed of a toxin protein that inhibits bacterial growth and an antitoxin that can protect bacteria from toxins. While the toxin (most often a protein) is stable, the antitoxin (either a protein or an RNA) is labile and can be easily degraded. One of the main features of bacterial persistence is a low metabolic state that may arise due to toxin activity. Antibiotics and other stress conditions can unbalance the relationship between the toxin and antitoxin, favouring the amount of toxins that lead bacteria to a transient dormancy state, which allows resistance against a high concentration of most antibiotics [6]. Persistence represents a tough obstacle to antibiotics because these molecules normally require strong metabolically active bacteria to exert their antibacterial effects. There are eight different types of TA systems depending on the antitoxin chemical nature and its mode of action; however, Type I and Type II are better characterized for *Escherichia coli* and *Pseudomonas aeruginosa*. Some toxins related to bacterial persistence have been identified: HokB, HipB, RelE, MazF YafQ, MqsR and DinJ [7]. Much research is focused on examining peptides with the ability to hyperactivate toxin activity under normal conditions in order to eradicate bacteria, and as such offer potential new anti-microbial agents [8]. However, at present, no new remedy active against bacterial toxin–antitoxin is ready for use.

### 3.2. Stringent Response and SOS Response

Stringent response is a conservative adaptation to nutrient starvation facilitating another mechanism of bacterial dormancy/persistence. The stringent response not only promotes bacteria to enter into a survival mode, it also enables bacteria to be prepared to utilize the lower amounts of nutrients available in the environment. Guanosine tetraphosphate (ppGpp) accumulation downregulates DNA replication and rRNA synthesis [9]. SOS response controls DNA repair through the factors LexA and RecA. Bacterial persistence is characterized by the maintenance of DNA repair. This molecular mechanism will be the base for future strategies for treating bacterial persistence, by either directly targeting the persistent bacterial cells or blocking their formation or resuscitation [10]. However, the natural resuscitation of bacteria after the conclusion of the antibiotic treatment suggests there could be a way to eliminate the persistent cells using a pulsed antibiotic course, even if more research is needed to indicate the best timing of pulsed therapy [10].

## 4. In Vivo Animal Studies

*Escherichia coli* is generally considered an extracellular bacterium; nevertheless, the uropathogen *E. coli* (UPEC) has lately been established as an opportunistic intracellular pathogen [11] in murine models (see Figure 2 below). UPEC has been demonstrated to proliferate swiftly, forming large inclusions called intracellular bacterial communities (IBCs) within the urothelium cells, from where *E. coli* can migrate out, ready to invade other urothelium cells, or persist quiescently for long periods undetected by the immune system and remain insusceptible to standard antibiotic therapies [12]. These studies show that in recurrent UTIs, the infection is not only extracellular in the urine, but also intracellular in some urothelial cells, which is the origin of subsequent UTIs. The whole IBC pathogenic cycle is well documented in mice, but little information is available about the pathogenic cycle in humans.

## 5. In Vivo Human Studies

Intracellular bacterial communities (IBCs) have long been reported in the urine of patients with acute cystitis [14], but they have not yet been detected in the urine of patients in between two acute episodes of rUTIs as proof of the existence of bacterial persistence and of a pathogenetic link between subsequent UTIs. The morphology of IBCs in the urine sediment (see Figure 3 below) is well characterized as illustrated by Eirnaes K [12]: the only problem seems to be not knowing how to describe and report it. This will be possible only if, in time, specific studies are conducted to look for IBCs in the urine of asymptomatic patients suffering from rUTIs [15,16].

## 6. Current Clinical Practise

Standard therapy for recurrent UTIs is to treat each episode with a short antibiotic administration according to the ongoing guidelines for “acute uncomplicated cystitis” [17,18,19] even if these guidelines are not specific for recurrent UTIs. Since the pathogen causing cystitis is the same most of the time, the patient may self-medicate with a prearranged prescription by one’s general practitioner. Unfortunately, antibiotic resistance is a common side effect of this strategy within a couple of years [20].

Alternatively, when relapses occur very close to one another, there can be the option of continuous prophylaxis. The use of long-term low-dose antibiotics (Nitrofurantoin 50 mg OD, Cephalexin 500 mg OD, or Cotrimoxazole 40/200 mg OD) decreases the rate of infection up to 95%. The antimicrobial choice is based on bacterial culture and antibiotic sensitivity testing, but there is no clear evidence for the optimal length of therapy, even if an initial six-month prescription is mostly used [21,22]. Regrettably, once prophylaxis ceases, patients often experience UTI relapse. It is not well known how many patients experience resolved rUTIs one year after ending six months of continuous prophylaxis. In a small trial with 15 women, it was reported that just a third of the patients experienced no recurrences in the year following the end of prophylaxis [23]. Women whose recurrent UTIs are associated with sexual intercourse should be offered postcoital prophylaxis. This involves taking a single dose of an effective antimicrobial (e.g., Nitrofurantoin 50 mg, Cotrimoxazole 40/200 mg, or Cephalexin 500 mg) after sexual intercourse [24]. Moreover, in 1993, Raz and Stamm [25] reported a 75% reduction in UTI in postmenopausal women with recurrent infections randomized to estrogen vaginal cream, resulting in reduced vaginal colonization of pathogenic bacteria and lowered vaginal pH.

Finally, there is the choice of managing recurrent UTIs without antimicrobials. Although studies show conflicting results, patients can be offered probiotics, cranberry tablets and D-Mannose [26].

## 7. Proposed Future Clinical Practise

Future management of recurrent UTIs should take into consideration the concept of IBCs and bacterial persistence so that there are specific protocols for recurrent UTI therapy, instead of using the guidelines designed for acute uncomplicated cystitis, where treatment is delivered as many times as needed.

The presence of intracellular bacterial communities (IBCs) in the urothelial cells implies the need for intracellularly active antimicrobials, so that it is possible to clear not only the bacteria present in the urine, but also those inside the mucosa of the bladder. This does not mean a change in treatment of acute uncomplicated UTIs, where the efficacy of treatment guidelines is unchallenged. The current research presented here [11,12] shows that in recurrent UTIs, the infection is also inside the urothelial cells, and not only in the urine, hence the need for intracellular active antibiotics. Infections due to intracellular bacteria are hard to treat due to the inability of many antimicrobial agents to penetrate mammalian cells.

### 7.1. Intracellular Active Antibiotics

Some antibiotics, such as macrolides, fluoroquinolones, tetracyclines, and rifampicin, are known to be active against intracellular organisms while others, such as beta-lactams and aminoglycosides, show poor intracellular activity. At the same time, fosfomycin and cotrimoxazole demonstrate intermediate intracellular activity [27]. Indeed, only some antibiotics with intracellular activity have a good spectrum against uropathogens. Therefore, macrolides, tetracyclines and rifampicin, with known intracellular activity, despite being important in the treatment of STDs, have no role in the therapy of rUTIs.

### 7.2. Pulsed Antibiotic Therapy

Finally, the concept of bacterial persistence should be addressed; the resuscitation of antibiotic-refractory dormant bacteria is achieved so that antibiotics can complete their action. Unfortunately, ongoing studies detailing the pathogenesis of bacterial persistence have not yet resulted in treatments to interfere with this phenomenon. At present, we remain with only one method of achieving the resuscitation of dormant bacteria, that is, by repeated interruptions of antibiotics. The application of pulsed antibiotic therapy has not been clarified by in vitro studies of bacterial persistence, i.e., the best optimum length of antibiotic suspension to achieve resuscitation of bacterial dormancy has not yet been established. However, as clinicians see patients with rUTIs, we propose a possible treatment for future use. If an acute UTI is treated with a 3–5-day antibiotic cycle, which is mostly insufficient for recurrent UTIs, a rational proposal would be to repeat, four times, the 3–5-day antibiotic cycle, 3–5 days apart (see Figure 4 below).

Ciprofloxacin 500 mg BID and Cotrimoxazole 160/800 BID would fit in the 3-day cycle, while Nitrofurantoin 100 mg BID and Pivmecillinam 400 mg TID would fit in the 5-day cycle. Fosfomycin 3 g OD stands alone in a 2-day administration. Intermittent antibiotic therapy is rarely prescribed in medicine; exceptions are pulmonary disease due to atypical or typical mycobacteria [28] and bronchiectasis as prophylaxis of acute exacerbation [29]. Moreover, recent laboratory experiments suggest caution in using intermittent antibiotic treatment due to the risk of favouring the rapid evolution of antimicrobial resistance [30,31]. The emergence of antibiotic-resistant bacteria is a problem mainly related to incorrect use of antibiotics and is more often seen in upper and lower respiratory tract infections with a viral etiology, where antibiotics should not be used at all [32]. In rUTIs, antibiotic resistance is not uncommon, and a patient can typically present after several failed therapies with Escherichia coli or Klebsiella pneumoniae ESBL, and resistance even to Cotrimoxazole and Ciprofloxacin. In this setting, therefore, the concern of increasing antimicrobial resistance by using pulsed antibiotic therapy cannot be denied.

### 7.3. Drug Combination

Drug combination, a strategy already employed in the treatment of febrile neutropenia, mycobacteria lung disease [28], and more recently *Helicobacter pylori* [33,34], could be the ideal option, even in recurrent UTIs. Ciprofloxacin, Cotrimoxazole and Fosfomycin could be the backbone of the combination therapy based on their intracellular activity, while Pivmecillinam, Nitrofurantoin or Gentamycin could be added to avoid bacterial resistance.

Other choices, like bacteriophage therapy, are possible, but not yet feasible. This fascinating option at the moment is limited to the Russian Federation, but is not yet used in the Western world due to a lack of Phase I, II and III clinical studies, which are standard processes before placing a new medication on our market [35]. Bacteriophages are viruses that cannot infect human cells, but attack and kill bacteria, both replicating and persistent.

### 7.4. Backbone and Ancillary Therapy

Therefore, backbone treatment options could be the following: Ciprofloxacin, 500 mg BID for 3 days a week for a month. Similarly, the same schedule could probably be used with Cotrimoxazole, 160/800 mg BID for 3 days a week and so on. Another possibility could be Fosfomycin 3 g on two consecutive days every week for a month. The association therapy should be used with any option based on antibiogram sensibility. Therefore, the ancillary drugs could be Pivmecillinam, Nitrofurantoin or Gentamycin regardless of their intracellular activity, since their function would be to protect the backbone therapy from losing its effectiveness.

Evidence-based medicine would require any new therapy to be compared with the standard of care, i.e., with continuous prophylaxis for 6 months with the same antibiotics at a low dosage (Ciprofloxacin 250 mg daily; Cotrimoxazole 80/400 mg daily, Fosfomycin 3 g weekly), and determining how many patients are free of rUTIs in the twelve months after treatment.

### 7.5. Back-Up Plan 

Since the use of pulsed antibiotic therapy has not been clarified by in vitro studies of bacterial persistence, and in the absence of an exact understanding of the best length of antibiotic suspension to achieve resuscitation of bacterial dormancy, it may be necessary to apply the same pulsed antibiotic therapy but with longer suspensions between the cycles. Instead of 4- to 5-day antibiotic suspensions, the same therapies could be working with 9- to 11-day interruptions on alternate weeks.

The authors are already examining their experiences based on the above concepts, which hopefully before long will be presented in a pilot study.

## 8. Conclusions

The purpose of this review about rUTIs is to suggest a new clinical perspective based on research regarding bacterial persistence and intracellular bacterial communities. Pulsed antibiotic therapy with intracellular active preparations represents a potentially novel concept that could be tested in clinical trials; pulsed Ciprofloxacin, Cotrimoxazole or Fosfomycin associated with Pivmecillinam, Nitrofurantoin or Gentamycin could be interesting new options for the treatment for rUTI, as clinicians feel they are left without treatment options.

Our aim is to inspire a controlled multicenter clinical trial of pulsed antibiotic therapy for rUTIs.

## Figures and Tables

**Figure 1 antibiotics-14-00022-f001:**
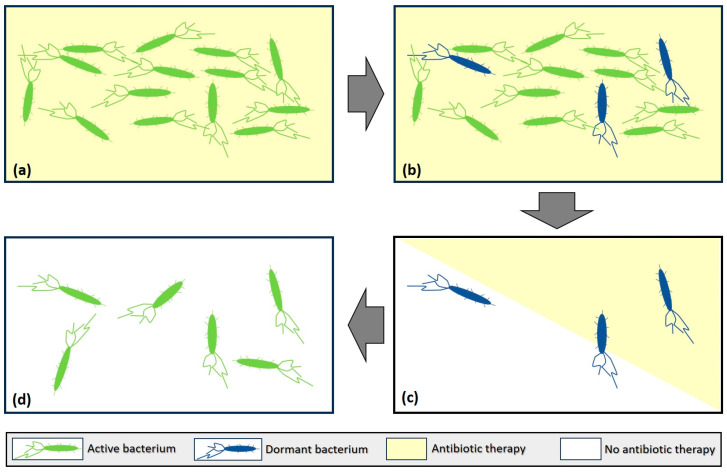
A bacterial population undergoing antibiotic therapy and the emergence of a fraction of persistent cells. Panel (**a**) shows the bacterial population undergoing antibiotic therapy, panel (**b**) represents emerging dormant bacteria under the antibiotic therapy, panel (**c**) shows the surviving dormant bacteria before becoming active again as indicated in panel (**d**).

**Figure 2 antibiotics-14-00022-f002:**
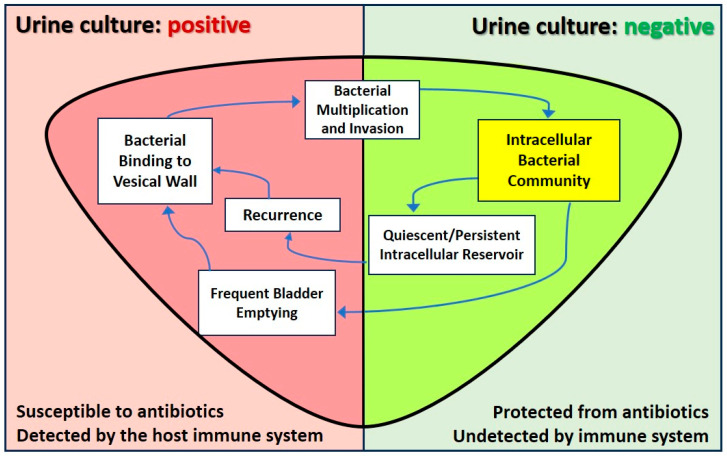
Diagram of the urinary bladder with intracellular bacterial community (IBC) maturation and development. The left half of the bladder (urine culture: positive) shows phases of IBC development during which bacteria are present in the lumen of the bladder with bacteria susceptible to antibiotics and detected by the host immune system. The right half of the bladder (urine culture: negative) shows phases of IBC development with bacteria protected from antibiotic therapy and undetected by the host immune system (modified from Kim A and others, 2021) [13].

**Figure 3 antibiotics-14-00022-f003:**
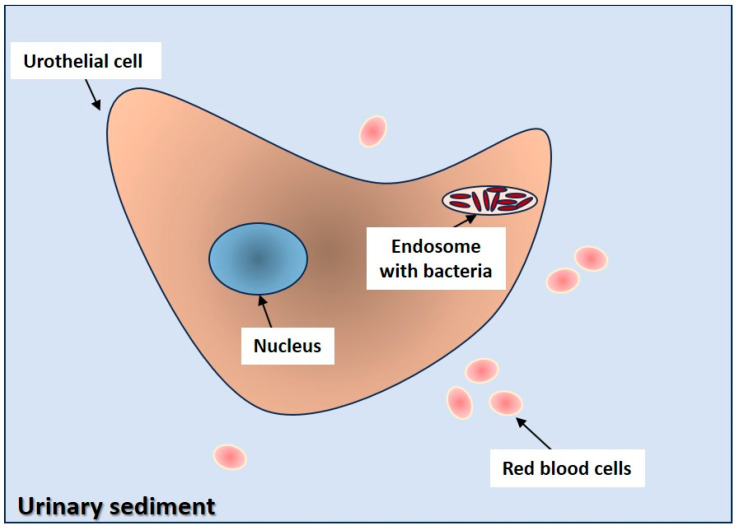
An illustration of IBCs of bacteria inside an endosome of a urothelial cell as may be found in the urinary sediment.

**Figure 4 antibiotics-14-00022-f004:**
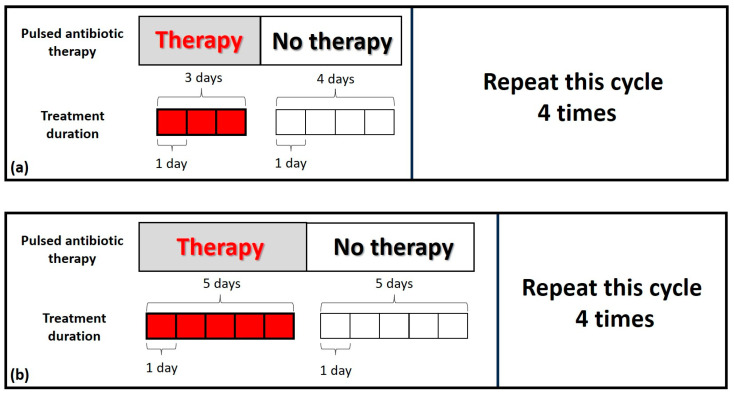
Pulsed antibiotic therapy with two different schedules: (**a**) the 3-day cycle with 4-day interruption; (**b**) the 5-day cycle with 5-day interruption.

## Data Availability

No new data were created or analyzed in this study.

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
