# Peer review of "Recurrent Urinary Tract Infections (UTIs): A Review and Proposal for Clinicians"

_antibiotics, 2025, doi:10.3390/antibiotics14010022_

Round 1
Reviewer 1 Report
Comments and Suggestions for Authors
This article proposes new therapies for the treatment of recurrent urinary tract infection. The review research performed is up-to-date and demonstrates the expertise of the authors. However, in order to be published, the authors could improve the quality of the images presented, especially Figure 2. In addition, the authors should include and discuss further studies in item 7. "Proposed future clinical" practice
Author Response
Comments 1: Thank you for your suggestions. Figure 2 has been completely redrawn and it looks now much better than before. Chapter 7. “Proposed future clinical practise” has been implemented by adding two paragraphs, as recommended.
Reviewer 2 Report
Comments and Suggestions for Authors
The authors provided a special and meaningful perspective of causes for rUTIs, an recommended some schemes, it is very suggestive for the practice. Could the authors provided more details about the actural effects of those schemes based on current infomation, and why to recommend them.
Author Response
Comments 2: The authors are already examining their experiences based on the concepts of bacterial persistence and the presence of intracellular bacterial communities (IBCs) into the urothelial cells which hopefully before long will be presented in a pilot study. This perspective paper does not provide any recommendation for treating rUTIs because this might come only after a controlled multicenter clinical trial of pulsed antibiotic therapy. However, we present what could be a pulsed antibiotic therapy with intracellular active preparations as novel concept that could be tested in future clinical trials.
Reviewer 3 Report
Comments and Suggestions for Authors
Recurrent urinary tract infections (UTIs) are a common health problem among the female population, and treatment can be difficult due to the emergence of bacterial resistance to antibiotics.
In the manuscript "Recurrent Urinary Tract Infections (UTIs): a review and proposal for clinicians" the authors reviewed the literature and proposed a treatment perspective based on the intracellular activity of antibiotics.
It would be interesting to know if the authors had results with this proposed treatment strategy.
Author Response
Comments 3: The authors are already examining their experiences based on the concepts presented in our paper which hopefully before long will be presented in a pilot study. This comment has been added at the of paragraph 7.